# COVID-19 Vaccine Hesitancy and Uptake among Minority Populations in Tennessee

**DOI:** 10.3390/vaccines11061073

**Published:** 2023-06-07

**Authors:** Donald J. Alcendor, Patricia Matthews-Juarez, Neely Williams, Derek Wilus, Mohammad Tabatabai, Esarrah Hopkins, Kirstyn George, Ashley H. Leon, Rafael Santiago, Arthur Lee, Duane Smoot, James E. K. Hildreth, Paul D. Juarez

**Affiliations:** 1Department of Microbiology, Immunology and Physiology, School of Medicine, Meharry Medical College, 1005 Dr. D.B. Todd Jr. Blvd., Hubbard Hospital, 5th Floor, Rm. 5025, Nashville, TN 37208, USA; 2Center for AIDS Health Disparities Research, Department of Microbiology, Immunology, and Physiology, School of Medicine, Meharry Medical College, 1005 Dr. D.B. Todd Jr. Blvd., Nashville, TN 37208, USA; 3Department of Family & Community Medicine, Meharry Medical College, 1005 D.B. Todd Jr. Blvd., Nashville, TN 37208, USA; 4Community Partners’ Network, Nashville, TN 37208, USAaleeart1@gmail.com (A.L.); 5School of Graduate Studies, Meharry Medical College, 1005 D.B. Todd Jr. Blvd., Nashville, TN 37208, USA; 6Division of Public Health, Meharry Medical College, 1005 D.B. Todd Jr. Blvd., Nashville, TN 37208, USA; 7Department of Internal Medicine, School of Medicine, Meharry Medical College, 1005 D.B. Todd Jr. Blvd., Nashville, TN 37208, USA

**Keywords:** COVID-19, vaccinations, Tennessee, vaccine hesitancy, minorities, vaccine uptake, disparities

## Abstract

COVID-19 vaccine hesitancy and uptake among Southern states in the US has been problematic throughout the pandemic. To characterize COVID-19 vaccine hesitancy and uptake among medically underserved communities in Tennessee. We surveyed 1482 individuals targeting minority communities in Tennessee from 2 October 2021 to 22 June 2022. Participants who indicated that they did not plan to receive or were unsure whether to receive the COVID-19 vaccine were considered vaccine-hesitant. Among participants, 79% had been vaccinated, with roughly 5.4% not likely at all to be vaccinated in the next three months from the date that the survey was conducted. When focusing particularly on Black/AA people and white people, our survey results revealed a significant association between race (Black/AA, white, or people of mixed Black/white ancestry) and vaccination status (vaccinated or unvaccinated) (*p*-value = 0.013). Approximately 79.1% of all participants received at least one dose of a COVID-19 vaccine. Individuals who were concerned with personal/family/community safety and/or wanted a return to normalcy were less likely to be hesitant. The study found that the major reasons cited for refusing the COVID-19 vaccines were distrust in vaccine safety, concerns about side effects, fear of needles, and vaccine efficacy.

## 1. Introduction

The World Health Organization reported vaccine hesitancy as one of the top 10 global concerns in 2019 [1]. A delay in or refusal of a vaccine, even if the vaccine is proven to be safe and effective and is widely accessible to the public, is defined as vaccine hesitancy [2,3]. Studies by Jafar et al., employing a single-arm meta-analysis, showed that the global prevalence of COVID-19 vaccine hesitancy is approximately 25% [4]. Various factors influence vaccine hesitancy. These factors pertain to age, ethnicity, gender, income levels, level of education, historical mistrust, access to health insurance, in biomedical and healthcare establishments, religious and moral convictions, political affiliation, and mistrust of the scientific community and medical providers [5]. A high level of hesitancy exists in medically underserved minority communities towards FDA-approved COVID-19 vaccines [6,7,8,9] that have been shown to be safe and effective [10]. In studies performed by Aw et al., vaccine hesitancy across high-income countries or regions ranged from 7–77.9%. Individuals who were younger, non-white, and had lower education levels were associated with increased vaccine hesitancy [11].

Long-term racial injustices and cultural insensitivity experienced by minority communities have supported a higher level of COVID-19 vaccine hesitancy, especially in Southern US, including Tennessee [12,13,14]. COVID-19 vaccine hesitancy fueled by misinformation has greatly influenced vaccine confidence and uptake in the US and around the world, consequently hindering efforts to mitigate the emergence of variants, infections, and viral transmission [15,16,17]. Misinformation has greatly contributed to vaccine hesitancy and has resulted in poor vaccine uptake.

The timely uptake of primary COVID-19 vaccines and boosters is necessary for reducing community spread and viral variant emergence [18]. Our community engagement efforts, vaccination events, and mobile vaccination programs targeted medically underserved minority communities in Tennessee. These efforts aimed to provide up-to-date, science-based information delivered by trusted messengers embedded in these communities to improve vaccine awareness, education, and vaccine uptake. We surveyed the participants at vaccine events to characterize COVID-19 vaccine hesitancy in these communities to enhance COVID-19 prevention efforts by reducing virus transmission and improving COVID-19 vaccine equity among medically underserved minorities in Tennessee. Tennessee follows the Centers for Disease Control and Prevention (CDC) policy guidelines which are executed via the Tennessee Departments of Health and healthcare providers. Medical care for COVID-19 in Tennessee is dependent on the types of insurance coverage patients have, along with government subsidies that are provided.

Knowledge gaps that we aim to address in this study are to identify the level of COVID-19 infections in this population, determine barriers to vaccine information and access, and to identify trusted messengers in these communities that can deliver evidence-based information about COVID-19 vaccine safety and efficacy to combat misinformation. 

## 2. Methods

**Surveys:** Exactly 1482 surveys were administered to participants that were 18 years and older and were among predominantly medically underserved minority communities in Tennessee at pop-up vaccine events, cultural gatherings, churches and other places of worship, pediatric clinics, metro parks, back-to-school events, community-based organization events, health department events, large and small businesses, festivals, nursing homes, and assisted-living facilities. The surveys took approximately 15–20 min to complete. Survey participants were asked to respond to the 27 questions from the Common Survey of the National Institutes of Health (NIH)-sponsored Tennessee Community Engagement Alliance against COVID-19 (TN CEAL). We administered most of the surveys in the urban counties in Tennessee that included Davidson, Shelby, Montgomery, Rutherford, Williamson, Sumner, Wilson, Robertson, Dickson, and Cheatham County. The questionnaire was designed and validated by the National Institutes of Health in the US. The inclusion criteria for participation in the survey were that participants had to reside in Tennessee and were 18 years and older. There were no exclusions if participants met these inclusion criteria. The survey aimed to access COVID-19 vaccine confidence vs. language, confidence vs. COVID testing, confidence vs. race, vaccine uptake vs. region, vaccine uptake vs. race, vaccine uptake vs. language, vaccine uptake vs. insurance, frequency and percentage of vaccine vs. COVID testing, frequency and percentage of vaccine vs. gender, and frequency and percentage of vaccine vs. education. **COVID-19 vaccination:** COVID-19 vaccination was provided via mobile vaccination programs in collaboration with the pediatric vaccine clinic of Meharry Medical College, the Tennessee Department of Health, and vaccine strike teams from the Vanderbilt School of Nursing. Pfizer/BNT and Moderna vaccines were administered by certified personnel following CDC guidelines. All participants receiving vaccines were registered. IRB-approved, de-identified vaccine information was included in the Survey Monkey database for analysis.


**Sample size calculations.**


The survey was completed by 1482 individuals, which is the sample size. Reporting a required minimum sample size for each item comparison made does not fall in line with the aims of the paper. Based on the 10 tables we made in the paper, the power ranges from: 0.254 to 0.999, 5 of which are about 90% or above.


**Multivariate analysis.**


The focus of this paper is to look at bivariate associations between variables in our survey. The use of the Chi-square and simulated Fisher’s exact tests, as we have described in our manuscript, are sufficient to answer these hypotheses.

*p*-values were added to the tables at the end of the manuscript. We believe we have provided sufficient information in the methodology section; the authors have no additional information to provide.

**Survey analysis**: Both paper and electronic surveys were administered at the vaccine events by TN CEAL staff, public health community ambassadors, and community healthcare workers. Paper surveys were sent to data monitoring staff who converted the collected information into Excel spreadsheets to be analyzed by a biostatistician. Continuous variables were analyzed using descriptive statistics such as mean, standard deviation, and 95% confidence intervals. Frequency and percentage analyses were performed on all binary and categorical survey questions. In addition, crosstabs were used to examine the association between pairs of categorical variables. Pearson’s Chi-square and, if necessary, Fisher’s Exact tests were performed to evaluate the significance of associations between these variables. Visual graphics were used to summarize the survey data. IBM SPSS Version 28.0 and R/RStudio were used to conduct all graphics and statistical analyses [19,20]. COVID-19 vaccine hesitancy is defined to participants as a reluctance/refusal to be injected with a COVID-19 vaccine after knowing that the vaccines via clinical trials have been proven to be safe and effective. It should be noted that in discussions with participants that were vaccine-hesitant and did not want the vaccines, they describe that their hesitancy was based on all available COVID-19 vaccines regardless of brand. 

**Ethical statement:** All subjects provided their informed consent for inclusion before they participated in the study. The study was reviewed by the Internal Review Board at Meharry Medical College and the IRB approval number is FWA00003675.

**STROBE Statement:** After a careful review of the 22 checklist of items that should be included in reports of observational studies in accordance with the STROBE guidelines, we conclude that the revised manuscript satisfies STROBE guideline requirements. 

## 3. Results

### 3.1. Assessment of Confidence Level in COVID-19 Vaccines among Survey Respondents

To gauge COVID-19 vaccine confidence in our target communities, we asked the participants how confident they were in the safety of COVID-19 vaccines currently available in the US. Figure 1 details how the participants responded based on the options provided. Among the 1482 participants, 637 (42.98%) were very confident in the safety of the COVID-19 vaccines currently available in the US, whereas 348 (23.48%) were somewhat confident. On the other hand, 136 participants (9.18%) were not too confident or not at all confident. A notable number of participants (261, 17.61%) had no response. Significant associations were found to be between confidence level and language (*p*-Value = 0.021), whether or not they received a COVID-19 test (*p*-Value = 0.027), and race (*p*-Value = 0.038) among those that responded to both questions, as seen in Table 1, Table 2 and Table 3. According to these tables, most individuals (48%) took the survey in English and were very confident in the effects of the vaccine. Among those that took the survey in Spanish, 65% were very confident in the vaccine; while 56% of those who took the survey in English were very confident. Within racial groups, 57% of Black people and 56% of white people were very confident; however, those who were Mixed were split with 41% reported being very confident and 40% reported being confident.

### 3.2. Assessment of Education Level, Gender, and Preferred Language of Survey Respondents

To ensure that our vaccine outreach included underserved communities in Tennessee, we assessed the racial demographics and ethnicity among the participants (Figure 2). Participants were allowed to select more than one race, as such 120 (8.10%) of all 1482 participants identified themselves as multiracial; this included 101 participants identifying with two races, 13 participants identified with three races, two participants identified with four races, and four participants identified with five races. Those that identified as multiracial were counted for each race category he or she identified with. Most of the participants surveyed identified as Black or Black and at least one other race (1077, 66.03%). The rest identified as white (197, 12.08%), Hispanic (265, 17.88%), American Indian or Alaskan Native (52, 3.19%), Asian (27, 1.66%), and Hawaiian (11, 0.67%) or a combination of that respective race. Ten (0.61%) refused to disclose their race and 257 (15.76%) did not respond. Hispanics were the only recorded ethnicity group, with a frequency of 265 (17.88%). We also examined the education levels, gender, and language preferences among survey participants (Figure 3A). Most of the participants (1135, 76.58%) completed at least a high school education. A total of 123 (8.30%) had some high school education, while 84 (5.67%) earned a GED. The rest either had less than a high school education (64, 4.32%), did not respond to this question (43, 2.90%), or preferred not to answer (33, 2.23%). Most of the participants were female (1072, 72.33%), while 389 (26.25%) were male and 21 (1.42%) were other (no response, prefer not to answer, transgender, or non-binary, genderqueer, genderfluid) (Figure 3B). The mean, standard deviation, and 95% confidence intervals for females and males were (mean = 49.31; SD = 16.85; 95% CI = (48.22, 50.49)) and (mean = 47.22; SD = 16.13; 95% CI = (45.48, 48.96)), respectively. Most of the participants selected English as their preferred language (1277, 86.17%). The rest selected Spanish (195, 13.16%) and 10 (0.67%) did not respond to the question (Figure 3C). 

### 3.3. Assessment of Employment and Health Insurance Coverage Status of Survey Respondents

To understand barriers to COVID-19 vaccine access among the medically underserved communities in Tennessee, we survey the employment and health insurance coverage status of the participants at the time of the study (Figure 4A). Roughly, one-third of the participants were working full-time (545, 36.77%). The rest of the participants selected the following options: retired (241, 16.26%), working for pay less than 40 h per week (204,13.77%), not able to work due to a disability (138, 9.31%), unemployed and looking for work (98, 6.61%), stayed at home (74, 4.99%), attending school (68, 4.59%), unemployed and not looking for work (23, 1.55%), on leave (layoffs) from their jobs (13, 0.88%), or working without pay (8, 0.54%) (Figure 4A). Most participants (1101, 74.29%) had either health insurance or a healthcare plan (Figure 4B). However, a significant number of participants (261, 17.61%) had no health insurance or healthcare plan (41, 2.77%), did not know if they had a plan and 79 (5.33%) did not respond to the question (Figure 4B).

### 3.4. Assessment of COVID-19 Testing Frequency and Positivity among Survey Respondents

To determine COVID-19 testing frequency and COVID-19 positivity rates, we asked the participants if they had ever been tested for COVID-19, the total number of times they had been tested during the survey period, and if they had ever tested positive for COVID-19 during the survey period. (Figure 5A). Our results show that most of the participants had been tested for COVID-19 (1175, 79.28%), while 248 (16.73%) were never tested and 59 (3.98%) did not respond to the question (Figure 5A). With regard to testing frequency, most of the participants had been tested for COVID-19 at least once (1148, 77.46%), while 334 (22.54%) did not respond to the question (Figure 5B). Finally, 339 (22.87%) of the participants reported testing positive for COVID-19 before, while 810 (54.66%) reported that they had not tested positive for COVID-19 (Figure 5C). On the other hand, 15 (1.01%) reported not knowing and 318, 21.46% did not respond to answer the question (Figure 5C). 

### 3.5. Assessment of COVID-19 Vaccine Uptake and Reasons for Vaccine Hesitancy

To determine vaccine uptake and identify reasons for vaccine hesitancy, we asked the participants about their experience with receiving COVID-19 vaccines and their reasons for being vaccine-hesitant. Our survey results show that over half of the participants (805, 54.32%) had received the complete primary vaccine series (both doses of the two-dose vaccine) (Figure 6A). For the rest of the participants, 241 (16.26%) received a single dose of the two-dose series with plans to have the second dose, 171 (11.54%) were unvaccinated, 126 (8.50%) received a one-dose vaccine, 75 (5.06%) provided no response, 45 (3.04%) preferred not to answer, and 19 (1.28%) did not know if they had been vaccinated for COVID-19 (Figure 6A). Significant associations were found between vaccine uptake and region (*p*-Value = 0.015), race (*p*-Value = 0.001), language (*p*-Value < 0.001), insurance (*p*-Value = 0.001), whether or not the individual was tested for COVID-19 (*p*-Value < 0.001), sex (*p*-Value < 0.001), and education (*p*-Value < 0.001) among those that responded to both questions, seen in Table 4, Table 5, Table 6, Table 7, Table 8, Table 9 and Table 10. When looking at sex, the estimated probability males got the one-dose vaccine (14.7%) was roughly twice as high as females (7.4%). When asked why they chose not to have the COVID-19 vaccine, most of the participants who had received the vaccine (1273, 85.90%) did not respond to this question (Figure 6B). Figure 6B also details how the rest of the participants responded based on the options provided. These responses indicate the main reasons for refusing the COVID-19 vaccines are distrust in vaccine safety and efficacy, concerns about side effects, and fear of needles (248, 16.73%). Some participants (28, 1.89%) refused to vaccinate because they did not believe themselves to be at risk of catching COVID-19. Some refused to be vaccinated because they felt that being injected with the vaccine conflicted with their religious beliefs (25, 1.69%), were concerned about being infected with COVID-19 at a vaccination event (16, 1.08%), their family or community did not approve of the COVID-19 vaccine (13, 0.88%), concerned about having to provide identification at a vaccination appointment (7, 0.47%), or felt that vaccination is not necessary since they had already had COVID-19 (6, 0.40%). A total of 28 participants (1.89%) had other reasons for not being injected with the vaccine (Figure 6B). 

### 3.6. Assessment of the Level of Trust in COVID-19 Information Sources

We asked the participants which entity they considered to be a reliable source of COVID-19 information based on the options we provided (Figure 7). Almost half of the participants (655, 44.20%) identified the Centers for Disease Control (CDC) as a very reliable source of COVID-19 information, while 374 (25.24%) identified the CDC as somewhat reliable (Figure 7A). On the other hand, 204 (13.77%) identified the CDC as an unreliable source (Figure 7A). The rest either did not know (57, 3.85%), thought it did not apply (88, 5.94%), or did not respond (104, 7.02%) (Figure 7A). Over a third of the participants (521, 35.16%) identified the US federal government as a somewhat reliable source of COVID-19 information, while 391 (26.38%) found the federal government to be a very reliable source (Figure 7B). On the other hand, 310 (20.92%) identified the federal government as an unreliable source of COVID-19 information. The rest of the participants either did not know (48, 3.24%), thought it did not apply (98, 6.61%), or did not respond (114, 7.69%) (Figure 7B). Almost a third of the participants (457, 30.84%) found the US Food and Drug Administration (FDA) to be a somewhat reliable source of COVID-19 information, while 415 (28.00%) found the FDA to be very reliable (Figure 7C). Almost a third of the participants (434, 29.28%) found the FDA to be unreliable (Figure 7C). The rest of the participants either did not know (123, 8.30%) or did not respond (53, 3.58%) (Figure 7C). More than half the participants (902, 60.86%) considered doctors and healthcare providers to be a very reliable source of COVID-19 information, while 222 (14.98%) considered them to be somewhat reliable. A total of 91 participants (6.14%) found doctors and medical providers to be unreliable (Figure 7D). The rest of the participants either did not know (87, 5.87%), thought it did not apply (78, 5.26%), or did not respond (102, 6.88%) (Figure 7D). Over a third of the participants (564, 38.06%) identified state and local governments as a somewhat reliable source of COVID-19 information, while 361 (24.36%) identified state and local governments as a very reliable source (Figure 7E). On the other hand, 292 (19.70%) of the participants found state and local government to be unreliable (Figure 7E). The rest of the participants either did not know (48, 3.24%), thought it did not apply (98, 6.61%), or did not respond (119, 8.03%) (Figure 7E). Almost half of the respondents (691, 46.63%) identified news on the radio, TV, online, or in newspapers as a somewhat reliable source of COVID-19 information, while 273 (18.42%) found these sources to be very reliable (Figure 7F). On the other hand, 267 (18.02%) found news on the radio, TV, online, or in newspapers to be unreliable (Figure 7F). The rest of the participants either did not know (42, 2.83%), thought it did not apply (96, 6.48%), or did not respond (113, 7.62%) (Figure 7F). Over a third of the participants (525, 35.43%) identified colleagues, classmates, or people they know as a somewhat reliable source of COVID-19 information, while 313 (21.12%) identified them as a very reliable source (Figure 7G). On the other hand, 295 (19.91%) of the respondents identified colleagues, classmates, or people they know as unreliable sources of COVID-19 information (Figure 7G). The rest of the participants either did not know (118, 7.96%), thought it did not apply (103, 6.95%), or did not respond (128, 8.64%) (Figure 7G). Over a third of the respondents (627, 42.31%) identified faith leaders as a very reliable source of COVID-19 information, while 263 (17.75%) identified faith leaders as a somewhat reliable source (Figure 7H). On the other hand, 202 (13.63%) of the respondents identified faith leaders as an unreliable source (Figure 7H). The rest of the participants either did not know (186, 12.55%), thought it did not apply (89, 6.01%), or did not respond (115, 7.76%) (Figure 7H). The largest number of respondents (902, 60.86%) identified primary care providers as a very reliable source of COVID-19 information.

## 4. Conclusions and Discussion

In the US, Tennessee ranks ninth for uninsured persons compared to other states [21]. The Kaiser Family Foundation has reported that more than 11% of Tennesseans, approximately 800,000 residents, are uninsured [22]. In the US, Tennessee had the fourth highest age-adjusted death rate per 100,000, due to COVID-19, among the 17 Southern states during the third quarter of 2022; this is the most recent data reported by the CDC [23]. Tennessee ranks eighth in the US in the COVID-19 death rates [24]. The 17 classified Southern states in the US consistently rank among the worst for health and wellness in the US. Southern states have the highest rates of premature deaths due to chronic conditions that can go untreated due to poverty, health disparities, and institutionalized racism compared to other states in the US [25,26,27]. Rates of obesity are higher in Southern states and individuals are more likely to smoke and live a sedentary lifestyle [28]. There is less access to healthcare providers in the South compared to other regions in the US [29]. The high incidence of chronic diseases found in the South is associated with comorbidities that predispose individuals to the most severe complications of COVID-19.

There is a significant level of vaccine hesitancy in Tennessee based on misinformation that would greatly impact vaccine confidence and uptake among medically underserved communities and that if we could engage these communities with correct information and free access to COVID-19 vaccines, we could improve vaccine confidence and identify barriers to vaccine uptake. A supermajority of respondents had confidence in vaccine safety. A vast majority of respondents were fully vaccinated. Those who had not been vaccinated cited distrust in vaccine safety and efficacy as a major concern. Nonetheless, most of the respondents were confident in the safety of the COVID-19 vaccines and placed a great deal of trust in the CDC, their healthcare providers, and their faith leaders. Over 79% of respondents tested for COVID-19, of whom 22.87% tested positive. Over three-fourths of respondents had at least one test for COVID-19. The survey supported our hypothesis in that if you understand barriers to vaccine uptake regarding the need for reliable science-based information, health coverage, and identifying trusted messengers, you can begin to expand these strategies to other communities in Tennessee to improve vaccine confidence and reduce hesitancy.

There is a disproportionately higher risk of mortality among minority populations with COVID-19 when compared to non-Hispanic white people [30,31,32]. Underlying clinical comorbidities result in poor clinical outcomes and increased mortality risk. Medically underserved populations in Tennessee experiencing poverty with little-to-no access to social services are more likely to have underlying medical conditions; hence, they are more likely to develop severe COVID-19 disease. These populations also have limited access to COVID-19 vaccines. In addition, these populations have been overburdened with myths and misinformation that contribute to vaccine hesitancy. Through initiatives sponsored by Meharry Medical College and supported by the Bloomberg Greenwood Initiative and the NIH-funded Tennessee Community Engage Alliance against COVID-19 (TN CEAL) grant, we continue to engage these communities in Tennessee in innovative ways to improve vaccine awareness, education, and access. We have engaged historically Black colleges and universities to recruit vaccine ambassadors and volunteers to promote evidence-based information to debunk myths and misinformation for communities in Tennessee. We have held vaccine events in collaboration with the Tennessee Departments of Health along with wrap-around and supportive care services that include free groceries, health insurance information, and services to improve pregnancy care for minority mothers in Tennessee. We have also engaged nursing homes and living-assisted facilities to ensure access to COVID-19 vaccines for those populations that are disabled, homebound, have limited mobility, and are most vulnerable to severe COVID-19 disease. We have collaborated with both small and large businesses throughout Tennessee to help vaccinate their workforce and provide an educational and awareness component to improve vaccine confidence. We routinely engage faith-based communities to host vaccine events for medical unreserved communities in Tennessee. Partnerships with faith-based communities allow us to interact with parents and children. We are aware that COVID-19 vaccination of children that require parental consent is challenging but in the context of faith-based community interactions, we have had marginal success. We have also attempted to engage parent–teacher associations (PTA) in Tennessee to educate parents about the safety and efficacy of the children’s COVID-19 vaccines. In Tennessee, we have a significant immigrant population that requires COVID-19 vaccination regardless of status. We have responded by vaccinating all eligible community members, regardless of immigration status.

Our goal in this study was to identify barriers observed by minority populations in Tennessee that could influence vaccine confidence and uptake and reduce vaccine hesitancy. To better understand barriers faced by Tennesseans, we have conducted statewide surveys and vaccination campaigns to improve vaccine confidence and vaccine equity among underserved communities. COVID-19 health inequities are rooted in systemic barriers that need to be addressed through community partnerships that can produce lasting improvements in public health for the underserved. At Meharry Medical College, we have developed collaborations with mobile vaccine programs, health departments, nursing schools at area universities, and community-based organizations to provide a network of community health workers and public health ambassadors in Tennessee to support public health and wellness.

Finally, on 11 May 2023, the COVID-19 associate National and Public Health Emergency declarations will end. The COVID-19 National Emergency and Public Health Emergency Declarations that have been ongoing since early January 2020 under the Trump administration and later extended under the Biden administration was a direct response to the COVID-19 pandemic that was declared by World Health Organization (WHO) on 11 May 2020. The declarations have provided COVID-19 vaccines and boosters, COVID testing, COVID treatments, as well as other services such as telemedicine [33]. Once these declarations expire, the financial burden will disproportionately impact underserved communities in Tennessee. The cost burden of vaccines will be placed on uninsured and underinsured populations in the US. In 2021, according to America’s Health Rankings, published by the United Health Foundation revealed that 10.1% of Tennesseans were uninsured compared to the national rate of 9.2% [34]. With a current population of 7.05 million, 712,050 Tennesseans are without health insurance. This number does not include the number of underinsured people in Tennessee. Tennessee is ranked 37th among the 50 states for the percentage of the population that is uninsured [35]. Affordability is the most important reason why Tennesseans are unable to obtain health insurance. The projected out-of-pocket expenses for COVID-19-related services among uninsured and underinsured populations in Tennessee will likely lead to widespread financial hardships. This will exacerbate conditions on the ground during periods of heightened community spread and increased infection rates that will likely increase the number of hospitalizations and deaths over time in Tennessee. The Public Health declarations over the years have greatly curtailed the pandemic in the US since the peak of the Omicron surge at the end of January 2022. As of 9 February 2023, nearly 270 million Americans have received at least one dose of a COVID-19 vaccine, daily COVID-19 reported cases are down 92%, COVID-19 deaths have declined by over 80%, and new COVID-19 hospitalizations are down nearly 80% [36].

Limitations of this study included that most surveys were administered in Middle Tennessee when compared to Western and Eastern Tennessee where there are higher levels of vaccine resistance and vaccine hesitancy. A significant number of respondents preferred not to answer some questions in the survey. The level and influence of misinformation among participants are unclear in the survey. The surveys administered were restricted to the study period (2 October 2021–22 June 2022). Some of the questions may have been confusing to participants, which tended to increase the non-response rate. The survey allowed the participants to select all options that apply, and it is plausible that some of the respondents overlapped in some of these responses. Participants may perceive the questions as too direct or judgmental and they simply will refuse to answer. We can improve our response rates in the future by simplifying survey questions and reducing the number of questions and question answers. We can also engage participants at vaccine events to encourage them to respond to all questions and stress the importance of fully completing the survey.

Policy makers should support legislation that provides a statewide campaign for COVID-19 vaccine awareness and education and promote the safety and efficacy of the booster to improve vaccine confidence and uptake in Tennessee. They should support free COVID-19 testing and vaccinations for communities in Tennessee that are uninsured after 11 May 2023 when the National and Public Health emergency declarations end.

Key implications of this study are that distrust in vaccine safety, concerns about side effects, fear of needles, and vaccine efficacy represents a common theme regarding the refusal of COVID-19 vaccine uptake in underserved populations. This theme is often supported by misinformation and historical distrust in the healthcare system.

## Figures and Tables

**Figure 1 vaccines-11-01073-f001:**
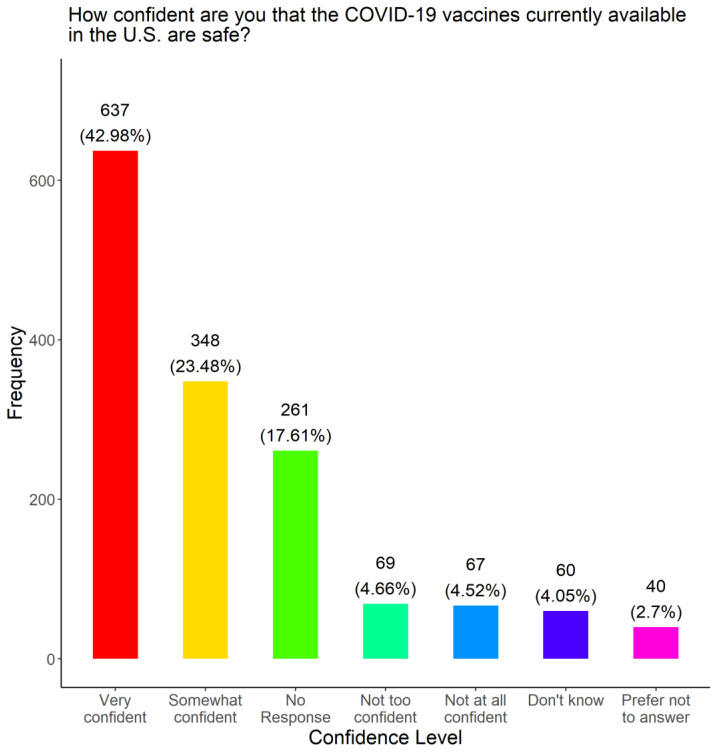
Confidence levels of survey respondents in COVID-19 vaccine safety.

**Figure 2 vaccines-11-01073-f002:**
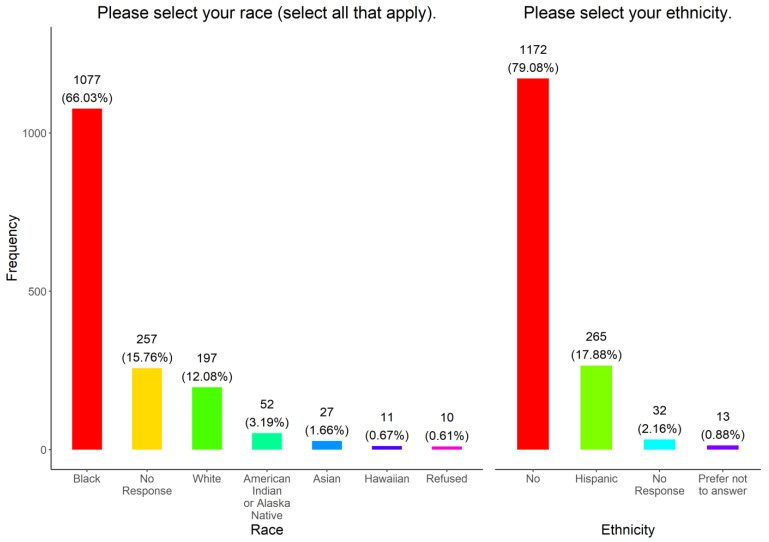
Race and ethnicity of survey respondents.

**Figure 3 vaccines-11-01073-f003:**
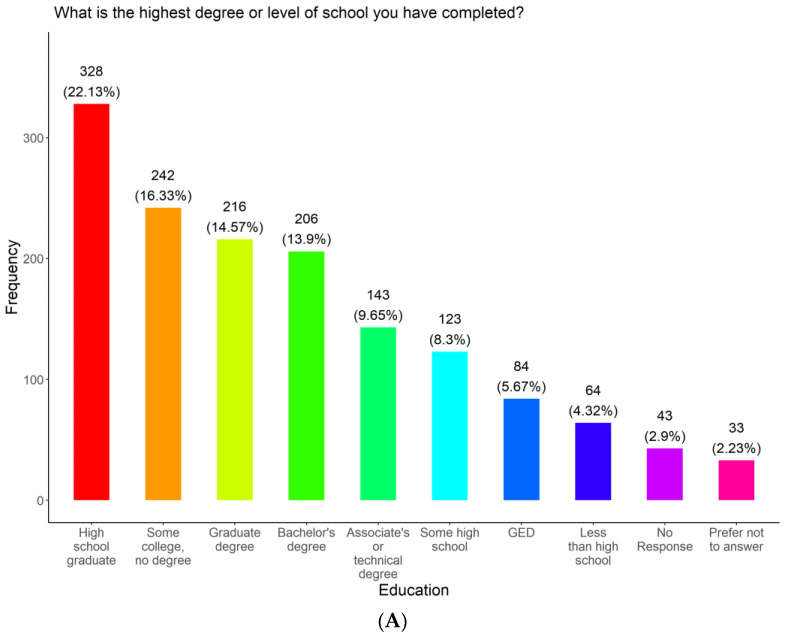
Education level, gender, and preferred language of survey respondents.

**Figure 4 vaccines-11-01073-f004:**
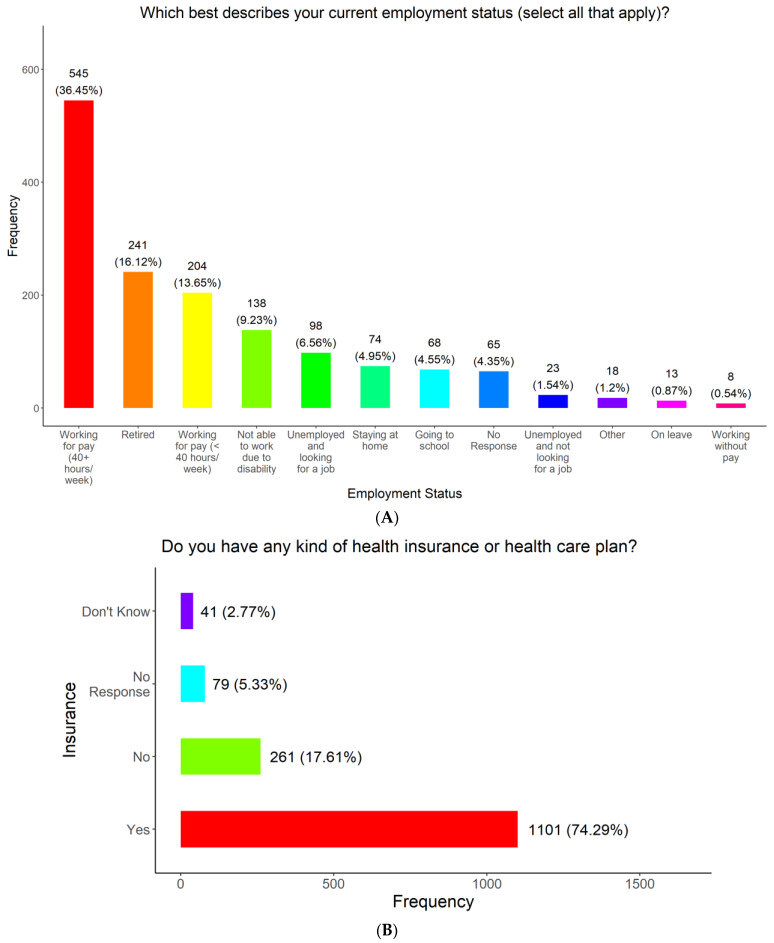
Employment and health insurance coverage status of survey respondents.

**Figure 5 vaccines-11-01073-f005:**
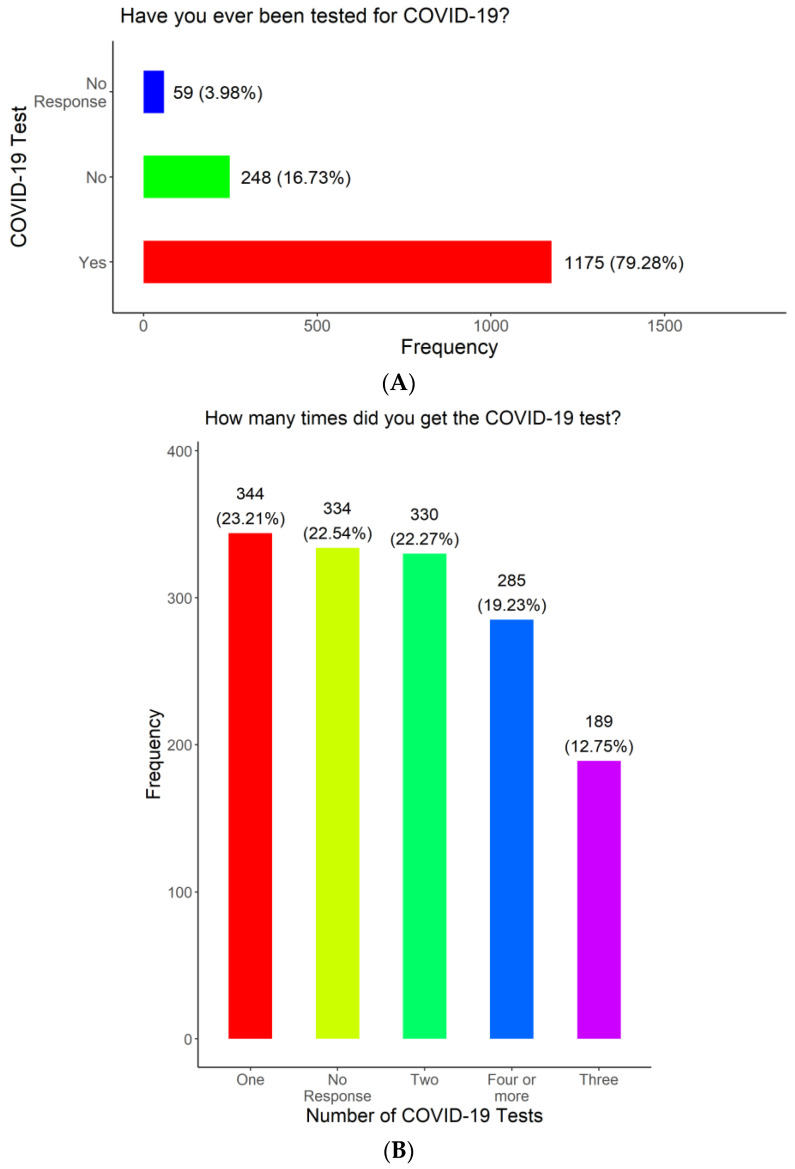
COVID-19 testing, testing frequency, and COVID-19 test positivity among survey respondents.

**Figure 6 vaccines-11-01073-f006:**
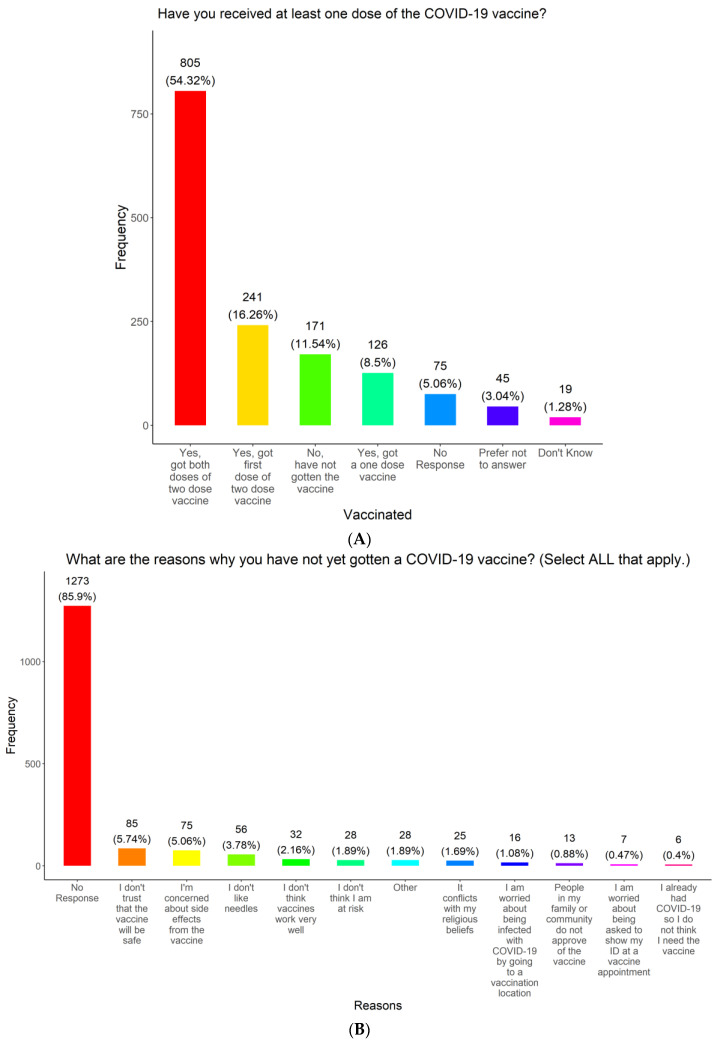
Vaccine uptake and reason for vaccine hesitancy of survey respondents.

**Figure 7 vaccines-11-01073-f007:**
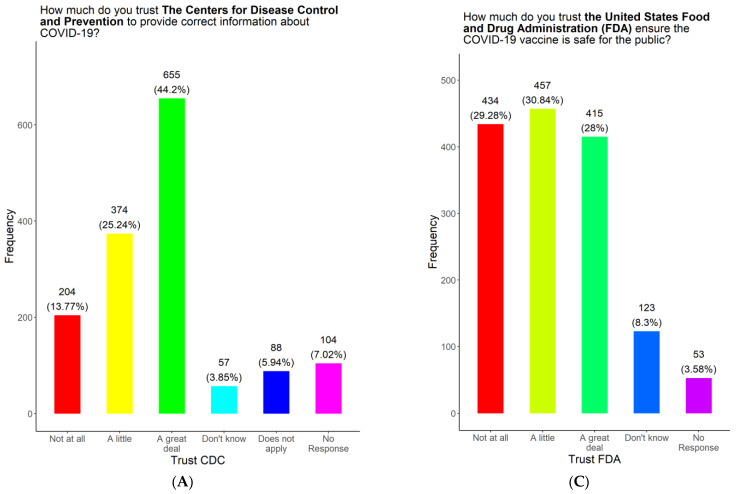
Trust levels in sources of COVID-19 information among survey respondents.

**Table 1 vaccines-11-01073-t001:** Frequency and Percentage of Confidence vs. Language.

		Language	
		English	Spanish
Confidence	Very Confident	537 (48%)	97 (9%)
	Somewhat Confident	303 (27%)	43 (4%)
	Not too Confident	62 (6%)	7 (1%)
	Not at all Confident	63 (6%)	2 (0%)

**Table 2 vaccines-11-01073-t002:** Frequency and Percentage of Confidence vs. COVID Testing.

		COVID Test	
		No	Yes
Confidence	Very Confident	87 (8%)	540 (49%)
	Somewhat Confident	60 (5%)	276 (25%)
	Not too Confident	11 (1%)	57 (5%)
	Not at all Confident	18 (2%)	48 (4%)

**Table 3 vaccines-11-01073-t003:** Frequency and Percentage of Confidence vs. Race.

		Race		
		Black	White	Mixed
Confidence	Very Confident	419 (46%)	48 (5%)	35 (4%)
	Somewhat Confident	230 (25%)	22 (2%)	34 (4%)
	Not too Confident	46 (5%)	4 (0%)	10 (1%)
	Not at all Confident	42 (5%)	9 (1%)	7 (1%)

**Table 4 vaccines-11-01073-t004:** Frequency and percentage of vaccine vs. region.

		Region		
		West	Central	East
Vaccine	Have not gotten the vaccine	69 (6%)	55 (5%)	29 (2%)
	Got a one-dose vaccine	54 (5%)	36 (3%)	16 (1%)
	Got first dose of two-dose vaccine	83 (7%)	85 (7%)	30 (3%)
	Got both doses of two-dose vaccine	381 (32%)	281 (24%)	74 (6%)

**Table 5 vaccines-11-01073-t005:** Frequency and percentage of vaccine vs. race.

		Race		
		Black	White	Mixed
Vaccine	Have not gotten the vaccine	108 (10%)	19 (2%)	17 (2%)
	Got a one-dose vaccine	67 (6%)	14 (1%)	9 (1%)
	Got first dose of two-dose vaccine	168 (15%)	12 (1%)	29 (3%)
	Got both doses of two-dose vaccine	560 (51%)	52 (5%)	47 (4%)

**Table 6 vaccines-11-01073-t006:** Frequency and percentage of vaccine vs. language.

		Language	
		English	Spanish
Vaccine	Have not gotten the vaccine	151 (11%)	20 (2%)
	Got a one-dose vaccine	97 (7%)	28 (2%)
	Got first dose of two-dose vaccine	221 (17%)	16 (1%)
	Got both doses of two-dose vaccine	701 (53%)	99 (7%)

**Table 7 vaccines-11-01073-t007:** Frequency and percentage of vaccine vs. insurance.

		Insurance	
		No	Yes
Vaccine	Have not gotten the vaccine	41 (3%)	119 (9%)
	Got a one-dose vaccine	32 (3%)	82 (6%)
	Got first dose of two-dose vaccine	37 (3%)	185 (15%)
	Got both doses of two-dose vaccine	113 (9%)	653 (52%)

**Table 8 vaccines-11-01073-t008:** Frequency and percentage of vaccine vs. COVID-19 test.

		COVID Test	
		No	Yes
Vaccine	Have not gotten the vaccine	49 (4%)	121 (9%)
	Got a one-dose vaccine	13 (1%)	110 (8%%)
	Got first dose of two-dose vaccine	31 (2%)	207 (16%)
	Got both doses of two-dose vaccine	123 (9%)	668 (51%)

**Table 9 vaccines-11-01073-t009:** Frequency and percentage of vaccine vs. gender.

		Sex	
		Male	Female
Vaccine	Have not gotten the vaccine	55 (4%)	114 (9%)
	Got a one-dose vaccine	50 (4%)	74 (6%)
	Got first dose of two-dose vaccine	62 (5%)	178 (13%)
	Got both doses of two-dose vaccine	173 (13%)	629 (47%)

**Table 10 vaccines-11-01073-t010:** Frequency and percentage of vaccine vs. education.

		Education							
		<HS	Some HS	HS	GED	Some College	Ass. or Tech.	Bachelor	Graduate
Vaccine	Have not gotten the vaccine	6 (0%)	27 (2%)	42 (3%)	20 (2%)	32 (2%)	14 (1%)	17 (1%)	8 (1%)
	Got a one-dose vaccine	10 (1%)	17 (1%)	33 (3%)	9 (1%)	11 (1%)	6 (0%)	17 (1%)	14 (1%)
	Got first dose of two-dose vaccine	9 (1%)	20 (2%)	71 (5%)	11 (1%)	42 (3%)	19 (1%)	35 (3%)	30 (2%)
	Got both doses of two-dose vaccine	27 (2%)	46 (4%)	146 (11%)	33 (3%)	145 (11%)	97 (7%)	129 (10%)	158 (12%)

## Data Availability

This manuscript did not report any laboratory-based data.

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
