# Peer review of "COVID-19 Vaccine Hesitancy and Uptake among Minority Populations in Tennessee"

_vaccines, 2023, doi:10.3390/vaccines11061073_

Round 1

Reviewer 1 Report

Dear editor,

Thank you for the kind invitation to review this manuscript.

The authors describe covid-19 vaccine hesitancy among underserved communities in Tenessee.

Below are my comments

Abstract

- the direction of the association should be highlighted 

- There is no conclusion in the abstract

Introduction 

- It will be helpful to highlight what are the key literature in the community in Tennesee or similar communities in US.

- More importantly, the knowledge gaps that the authors aim to target should be clearly written. 

Methods

- Why were 1482 surveys taken

-> What was the sample size for adequate power and the sample size computation

- How did the authors know if the participants were from medically underserved minority communities?

- The questionnaire and survey used should be described further - including the items and what they aim to assess. 

- Please report the guidelines in accordance to the STROBE checklist as there are multiple missing important information

- Is the questionnaire used validated? 

- The medical care and covid-19 related policies in Tennesse should be briefly mentioend for completeness

- What are the inclusion criteria of the study and exclusion criteria of the study?

- Was any pilot studies performed for the questionnaire given the authors highlighted significant issues with confusion about questions in the limitations 

- What is vaccine hesitancy defined as in the questionnaire 

- Please provide a copy of the questionnaire 

Results

- Suggest not to interwine methodology with results as it is confusing to follow

- p-value to report to max 4 significant figures

- please summarize the findings 

Discussion

- Specific sections e.g. demographics of residents in tennesse may have been better placed in the introduction

- The hypothesis should not be in the discussion

- There appears to be excessive discussion about vaccination programs in Tennesse without correlation with study findings

- What are the key implications of the study and things required moving forward 

- are there any unique findings from the study compared to other underprivilged populations in US and globablly

- How do the study results compare with those studies

- What should policy makers do to improve the vaccine hesitancy of residents in Tennesse 

Minor comments

- The high rates of vaccine hesitancy globally should be included in the introduction

Please cite the following relevant article: https://pubmed.ncbi.nlm.nih.gov/34452026/

Some grammatical errors and odd sentence phrasing 

Author Response

June 2, 2023

Editor and Chief
Journal Vaccines

Manuscript ID vaccines-2396802

Dear Editor,

My responses to reviewers’ comments regarding manuscript ID vaccines-2311161 entitled “COVID-19 Vaccine Hesitancy and Uptake among Minority Populations in Tennessee are enclosed.

Thank you for giving me the opportunity to resubmit this manuscript to your journal for publication.

Kind regards,

Donald J. Alcendor, Ph.D.

Associate Professor

Meharry Medical College

Center for AIDS Health Disparities Research

& Department of Microbiology and Immunology

& Obstetrics and Gynecology

Hubbard Hospital 5th Floor Rm. 5025

1005 Dr. D.B. Todd Jr. Blvd.

Nashville, TN 37208

Phone: 615-327-6449

Fax: 615-327-6929

Email: dalcendor@mmc.edu

Associate Professor Adjunct

Department of Pathology, Microbiology and Immunology

Vanderbilt University Medical Center

Comments and Suggestions for Authors

Reviewer #1

Abstract

- the direction of the association should be highlighted.

Authors’ response: It is unclear to us as to what the reviewer is alluding to regarding the direction of the association in the abstract.  There is no association listed in the abstract.

- There is no conclusion in the abstract

Authors’ response:  We somewhat disagree with the reviewer because the last sentence in the abstract is the conclusion from the results and the figures in the manuscript.

Introduction 

- It will be helpful to highlight what are the key literature in the community in Tennessee or similar communities in US.

Authors’ response:  We agree with the reviewer, however there is little to no information other than vaccination rates specific to Tennessee in the literature, therefore there is very little substantive data to compare and contrast.  We want to change this in the future.

- More importantly, the knowledge gaps that the authors aim to target should be clearly written. 

Authors’ response: We agree with the reviewer, and have provided information to the Introduction section of the revised manuscript in blue text.

Methods

- Why were 1482 surveys taken

Authors’ response: The 1482 surveys represents the number of surveys administered at vaccine events in Tennessee during the given time interval stated in the manuscript.  This number also represent surveys taken during a COVID surge in the US especially in Tennessee.

-> What was the sample size for adequate power and the sample size computation

Authors’ response: The power varies depending on the survey item. Although we would have liked to see a higher response rate to some survey questions, the rate is out of our control. Overall, the power is minimally sufficient.

- How did the authors know if the participants were from medically underserved minority communities?

Authors’ response:  We agree with the reviewer: This was established using geospatial data employing zip codes and analysis of race and socioeconomic data within specific zip codes in Tennessee.

- The questionnaire and survey used should be described further - including the items and what they aim to assess. 

Authors’ response:  We agree with the reviewer, and have added aims of the survey to the Methods section of the revised manuscript in blue text.

- Please report the guidelines in accordance to the STROBE checklist as there are multiple missing important information

Authors’ response: We somewhat disagree with the reviewer: We have added a STROBE statement to the revised manuscript in blue text. STROBE Statement: After careful review of the 22 checklist of items that should be included in reports of observational studies in accordance with the STROBE guide-line we conclude that the revised manuscript satisfies STROBE guideline requirements.  

- Is the questionnaire used validated? 

Authors’ response:  The questionnaire was designed and validated by the National Institutes of Health in the US. This information has been added to the Methods section of the revised manuscript in blue text.

- The medical care and covid-19 related policies in Tennessee should be briefly mentioned for completeness.

Authors’ response: We agree with the reviewer: Tennessee follows the Centers for Disease Control and Prevention (CDC) policy guidelines which are executed via the Tennessee Departments of Health and healthcare providers.  Medical care for COVID -19 in Tennessee is dependent on the types of insurance coverage patients have along with government subsidies that are provided. This information has been added to the Introduction section of the revised manuscript.

Authors’ response: 

- What are the inclusion criteria of the study and exclusion criteria of the study?

Authors’ response:  The inclusion criteria for participation in the survey was that participants had to reside in Tennessee and were 18 years and older. There were no exclusions if they met these inclusion criteria.  This information has been included in the Methods section of the revised manuscript in blue text.

- Was any pilot studies performed for the questionnaire given the authors highlighted significant issues with confusion about questions in the limitations 

Authors’ response:  The Common CEAL survey was piloted in communities by the NIH with input from the 21 CEAL teams in the US.

- What is vaccine hesitancy defined as in the questionnaire

Authors’ response:  We agree with the reviewer: vaccine hesitancy is defined to participants as a reluctance/refusal to get a COVID-19 vaccine after knowing that the vaccine via clinical trials have been proven to be safe and effective.  This information has been added in blue text in the Methods section of the revised manuscript. 

- Please provide a copy of the questionnaire 

Authors’ response:  The questionnaire was develop by the NIH specifically for the 21 CEAL Team groups from 21 states in the US.  Access to the CEAL Team common survey maybe accessible via the NIH CEAL server.

Results

- Suggest not to interwine methodology with results as it is confusing to follow

Authors’ response: We disagree with the reviewer, and have worked to organize the manuscript in a manner that is reader accessible.  The manuscript was reviewed by our editing department prior to submission.

- p-value to report to max 4 significant figures.

Authors’ response: P-values in the manuscript have been updated to show at most 4 significant digits.

- please summarize the findings 

Authors’ response:  The summary of the findings can be found in the conclusion section of the revised manuscript as outlined in blue text.

Discussion

- Specific sections e.g. demographics of residents in Tennessee may have been better placed in the introduction

Authors’ response: We somewhat disagree with the review in that the demographics of residents of Tennessee is well suited for the Results section of the manuscript.  

- The hypothesis should not be in the discussion

Authors’ response:  We agree with the reviewer and have reworded this section of the Discussion that appears in blue text of the revised manuscript.

- There appears to be excessive discussion about vaccination programs in Tennessee without correlation with study findings

Authors’ response: In Tennessee there are 95 counties and our vaccination program via the NIH Tennessee CEAL program is engage with vaccination programs that cover these 95 counties.  Our vaccine partnerships and our partnerships with community based organizations that provide wrap around services for participants that attend vaccine events are essential for the communities that we serve.

- What are the key implications of the study and things required moving forward 

Authors’ response:  We agree with the reviewer and have added information to the Conclusion section of the revised manuscript in blue text.

- are there any unique findings from the study compared to other underprivileged populations in US and globally

Authors’ response: The findings are not very unique from the majority of studies in underserved populations and we observe a common theme with other studies.   The theme of distrust in vaccine safety, concerns about side effects, fear of needles, and vaccine efficacy are considered common reasons for refusing the COVID-19 vaccines.  However, the majority of participants were fairly well educated and the majority has health insurance but the reasons were similar to other studies.

- How do the study results compare with those studies

Authors’ response:  Study results in underserved populations are very similar when identifying reasons for vaccine hesitancy. In addition, non-white populations with lower educations levels leverage by mistrust in the government and medical establishment is a common theme in these types of studies.

- What should policy makers do to improve the vaccine hesitancy of residents in Tennessee

Authors’ response:  I think the reviewer meant to reduce vaccine hesitancy in Tennessee. We agree with the reviewer and information has been added to the revised manuscript in blue text in the Conclusion section of the revised manuscript. 

Minor comments

- The high rates of vaccine hesitancy globally should be included in the introduction.

Authors’ response:  We agree with the reviewer and have included the rate of global COVID-19 vaccine hesitancy in the Introduction section in blue text of the revised manuscript along with the reference.

Please cite the following relevant article: https://pubmed.ncbi.nlm.nih.gov/34452026/

Authors’ response:  We agree with reviewer and added this reference and corresponding information to the Introduction section of the revised manuscript in blue text. The reference has been added to the reference section of the revised manuscript.

Reviewer 2 Report

1.     In the Method section:

(1)   Please indicate which brand of COVID-19 vaccine (e.g., AZ, Pfizer/BNT, Moderna, Novavax vaccine, etc.) is included in the surveys.

(2)    Please describe the sampling method to screen for the participants who administered COVID-19 vaccines in exactly 1482 surveys in details to show it statistical significance.

(3)   Please provide a Table to list the 27 questions from the Common Survey of the National Institutes of Health (NIH) sponsored Tennessee Community Engagement Alliance against COVID-19 (TN CEAL).

2.     In this manuscript, the authors did not evaluate and consider the influence and/or difference in administering different brand of COVID-19 vaccines for people in vaccine hesitancy.

3.     The results revealed a notable number of participants (261, 17.61%) had no response. Please discuss why some of the questions may have been confusing to participants, which tend to increase the non-response rate. Additionally, how to improve the response rate?

4.     In the Conclusion section:

(1)   The authors indicate that a significant number of respondents preferred not to answer some questions in the survey. Please explain how to show the data statistics is significant in this manuscript.

(2)   The abstract indicate that the study found that the major reasons cited for refusing the COVID-19 vaccines were distrust in vaccine safety, concerns about side effects, fear of needles, and vaccine efficacy. However, the authors did not show this point in this section.

5.     Please provide ethical statement and IRB approval number.

No.

Author Response

June 2, 2023

Editor and Chief
Journal Vaccines

Manuscript ID vaccines-2396802

Dear Editor,

My responses to reviewers’ comments regarding manuscript ID vaccines-2396802 entitled “COVID-19 Vaccine Hesitancy and Uptake among Minority Populations in Tennessee are enclosed.

Thank you for giving me the opportunity to resubmit this manuscript to your journal for publication.

Kind regards,

Donald J. Alcendor, Ph.D.

Associate Professor

Meharry Medical College

Center for AIDS Health Disparities Research

& Department of Microbiology and Immunology

& Obstetrics and Gynecology

Hubbard Hospital 5th Floor Rm. 5025

1005 Dr. D.B. Todd Jr. Blvd.

Nashville, TN 37208

Phone: 615-327-6449

Fax: 615-327-6929

Email: dalcendor@mmc.edu

Associate Professor Adjunct

Department of Pathology, Microbiology and Immunology

Vanderbilt University Medical Center

Reviewer #2

  1. In the Method section:

(1)   Please indicate which brand of COVID-19 vaccine (e.g., AZ, Pfizer/BNT, Moderna, Novavax vaccine, etc.) is included in the surveys.

Authors’ response: We agree with the review and have added Pfizer/BNT and Moderna as vaccine brands that were administered to participants in the methods section of the revised manuscript in blue text.

(2)    Please describe the sampling method to screen for the participants who administered COVID-19 vaccines in exactly 1482 surveys in details to show it statistical significance.

Authors’ response:  Tables were set up at various locations across Tennessee, specifically targeting urban locations. People were encouraged to participate in the survey through monetary incentives. Responses were tested using appropriate statistical tests and techniques. Results were then reported.

(3)   Please provide a Table to list the 27 questions from the Common Survey of the National Institutes of Health (NIH) sponsored Tennessee Community Engagement Alliance against COVID-19 (TN CEAL).

Authors’ response: We somewhat disagree with the reviewer because the table list of 27 questions with all possible responses would be 12 pages in length which is beyond the scope of this report.  The NIH CEAL Common Survey questions may be accessible on the NIH CEAL server.

  1. In this manuscript, the authors did not evaluate and consider the influence and/or difference in administering different brand of COVID-19 vaccines for people in vaccine hesitancy.

Authors’ response:  In discussions with participants that were vaccine hesitant, and did not want the vaccines they describe that their hesitancy was based on all available COVID vaccines regardless of brand This information has been added to the Methods section of the revised manuscript in blue text.

  1. The results revealed a notable number of participants (261, 17.61%) had no response. Please discuss why some of the questions may have been confusing to participants, which tend to increase the non-response rate. Additionally, how to improve the response rate?

The questions could be too wordy and complex for certain participants. The questions had many possible answers and may have confused participants.  Participants may perceive the question as too direct or judgmental and they simply will refuse to answer.  Participants could be signaling that they simply do not like the survey.  We can improve our response rates in the future by simplifying survey questions, reducing the number of questions and question answers. We can also engage participants at vaccine events to encourage them to respond to all questions and stress the importance of fully completing the survey. This information is included in blue text in the revised manuscript.

  1. In the Conclusion section:

(1)   The authors indicate that a significant number of respondents preferred not to answer some questions in the survey. Please explain how to show the data statistics is significant in this manuscript.

Authors’ response:  Appropriate statistical tests were used to determine the significance of the survey results.

  1. (2)The abstract indicate that the study found that the major reasons cited for refusing the COVID-19 vaccines were distrust in vaccine safety, concerns about side effects, fear of needles, and vaccine efficacy. However, the authors did not show this point in this section.

Authors’ response:  These findings are outlined in the results sections of the manuscript, the figures, and the discussion section of the manuscript.  This information is stated as a synopsis of the findings and are detailed in the abstract. 

  1. Please provide ethical statement and IRB approval number.

Authors’ response:  We agree with the reviewer and have included the IRB and the ethical statement in the revised manuscript in blue text.

Round 2

Reviewer 1 Report

nil further comments

-

Author Response

No response needed.

Reviewer 2 Report

The manuscript has been significantly improved.

No

Author Response

No response needed.

Donald J. Alcendor PhD